# Tempo and mode in karyotype evolution revealed by a probabilistic model incorporating both chromosome number and morphology

Kohta Yoshida [ORCID]¤, Jun Kitano [ORCID]*

Ecological Genetics Laboratory, National Institute of Genetics, Mishima, Japan

¤ Current address: Max Planck Institute for Developmental Biology, Tübingen, Germany
* jkitano@nig.ac.jp

## Abstract

Karyotype, including the chromosome and arm numbers, is a fundamental genetic characteristic of all organisms and has long been used as a species-diagnostic character. Additionally, karyotype evolution plays an important role in divergent adaptation and speciation. Centric fusion and fission change chromosome numbers, whereas the intra-chromosomal movement of the centromere, such as pericentric inversion, changes arm numbers. A probabilistic model simultaneously incorporating both chromosome and arm numbers has not been established. Here, we built a probabilistic model of karyotype evolution based on the "karyograph", which treats karyotype evolution as a walk on the two-dimensional space representing the chromosome and arm numbers. This model enables analysis of the stationary distribution with a stable karyotype for any given parameter. After evaluating their performance using simulated data, we applied our model to two large taxonomic groups of fish, Eurypterygii and series Otophysi, to perform maximum likelihood estimation of the transition rates and reconstruct the evolutionary history of karyotypes. The two taxa significantly differed in the evolution of arm number. The inclusion of speciation and extinction rates demonstrated possibly high extinction rates in species with karyotypes other than the most typical karyotype in both groups. Finally, we made a model including polyploidization rates and applied it to a small plant group. Thus, the use of this probabilistic model can contribute to a better understanding of tempo and mode in karyotype evolution and its possible role in speciation and extinction.

## Author summary

Karyotype, including chromosome number and morphology, has been observed even before DNA was identified as the primary genetic material. Although chromosomal changes are thought to play an important role in speciation, we have not reached a consensus on how rapidly karyotype can evolve and whether a particular karyotype is favored. This can be attributed partly to the lack of good probabilistic models for karyotype

**Data Availability Statement:** Karyotype data, R scripts and R results are deposited in the publicly accessible Dryad Digital Repository (https://

datadryad.org/stash/dataset/doi:10.5061/dryad.
s4mw6m966).

**Funding:** This research was supported in part by
Japan Society for the Promotion of Science (JSPS:
https://www.jsps.go.jp/) Kakenhi 17KT0028 and
19H01003 and JST CREST (https://www.jst.go.jp/
kisoken/crest/) JPMJCR20S2 to J.K. The funders
had no role in study design, data collection and
analysis, decision to publish, or preparation of the
manuscript.

**Competing interests:** The authors have declared
that no competing interests exist.

evolution. This contrasts with DNA sequence evolution, for which many established probabilistic models are available. Such probabilistic models have contributed to the understanding of rates and driving forces of DNA sequence evolution. Here, we built a probabilistic model including both chromosome and arm numbers. Using this model, we could demonstrate differences in the tempo and mode in karyotype evolution between two fish taxonomic groups and possible roles of karyotype in speciation and extinction. The use of our model in diverse taxa will lead to a better understanding of the evolutionary trends and functional roles of karyotypes.

## Introduction

The karyotype is a fundamental characteristic of all organisms [1–3]. For over a hundred years, the karyotype data, especially chromosome number and arm number (or fundamental number), has been collected from diverse taxa well before the beginning of the genomic era. Typically, karyotypes of closely related species often differ. Hence, it has long been used as a representative taxonomic character. Furthermore, karyotype evolution can play important roles in speciation. For instance, interbreeding of individuals with different karyotypes results in heterokaryotypic hybrids with impaired fertility, although there are cases that heterokaryotypes have normal fertility [4]. Therefore, karyotypic changes can contribute to genomic incompatibility [3,5]. Furthermore, divergence in karyotype can reduce recombination rates at rearranged regions in heterokaryotypes and contribute to the maintenance of divergent alleles involved in reproductive isolation between populations despite the presence of gene flow [6,7]. Therefore, elucidating the patterns of karyotype evolution and its association with speciation and extinction is essential for a better understanding of the evolution of biodiversity.

Analysis of the evolution of chromosome number has been conducted extensively. For example, the evolutionary rates of chromosome numbers were estimated by dividing the difference in chromosome number by species divergence time [8–10]. However, this calculation does not take into account reversible evolution such as fission followed by fusion and *vice versa*. More recently, model-based estimation using phylogenetic trees has been applied to evaluate chromosome number evolution [11–15]. These studies have demonstrated taxonomic differences in the rates of chromosome number evolution and a possible link between chromosome number evolution and speciation rate [14,15].

The majority of these previous studies, however, do not take chromosome morphology (*i.e.*, the position of the centromere) into account. Changes in chromosome morphology can be attributed to several mechanisms. First, a centric fusion of two acrocentric chromosomes, in which the centromeres are positioned at the periphery, leads to the formation of one metacentric chromosome, in which the centromere is positioned in the middle [16,17]. Second, the centric fission of a metacentric chromosome leads to the formation of two acrocentric chromosomes. Both of these translocations change the chromosome number, but not the arm number. Third, the movement of centromere within a chromosome can occur, for example, by pericentric inversion [18] and centromere repositioning [19,20]. In contrast to centric fusions and fissions, centromere movement modifies only the arm number and not the chromosome number.

Analysis of the evolutionary rate of chromosome morphology is important for several reasons. First, such analysis can provide insights into the evolution of female meiotic drive, because chromosome morphology is thought to be under female meiotic drive [2,3]. This idea is supported by the fact that chromosomes of particular morphology are preferentially

transmitted to the egg rather than the polar bodies in heterokaryotypes, and many mammalian species have karyotypes with all metacentric chromosomes or all acrocentric chromosomes [21]. Second, as chromosome morphology influences the evolution of chromosome numbers, we need to evaluate chromosome and arm numbers simultaneously for a better model of chromosome evolution. This can be attributed to the fact that acrocentric and metacentric chromosomes are materials for centric fusion and fission, respectively. Therefore, the number of acrocentric and metacentric chromosomes can affect the occurrence of centric fusion and fission, respectively. For instance, once all metacentric chromosomes undergo fission, the chromosome number does not increase in the absence of centromere movement. Centromere movement can generate metacentric chromosomes from acrocentric chromosomes to provide the opportunities for fission.

Here, we propose a probabilistic model incorporating both chromosome and arm numbers. Imai and Crozier proposed the use of "karyograph" to visualize a karyotype by plotting haploid arm number (*AN*) and haploid chromosome number (*n*) on the *x*-axis and *y*-axis, respectively [18]. Karyotype evolution can be simulated by a walk on the grids in the karyograph. Centric fission and fusion can be simulated by walking up and down, respectively, along the *y*-axis in the karyograph (Fig 1A and 1C). Centromere movement can be simulated by a horizontal walk along the x-axis in the karyograph (Fig 1B and 1C). Although Imai and colleagues summed up the difference in chromosome and arm numbers between the species being evaluated and used it as a karyotypic distance [22], they did not build any probabilistic model based on the karyograph. In the present study, we built a probabilistic model of karyotype evolution, based on a stochastic continuous time Markov process with the karyograph.

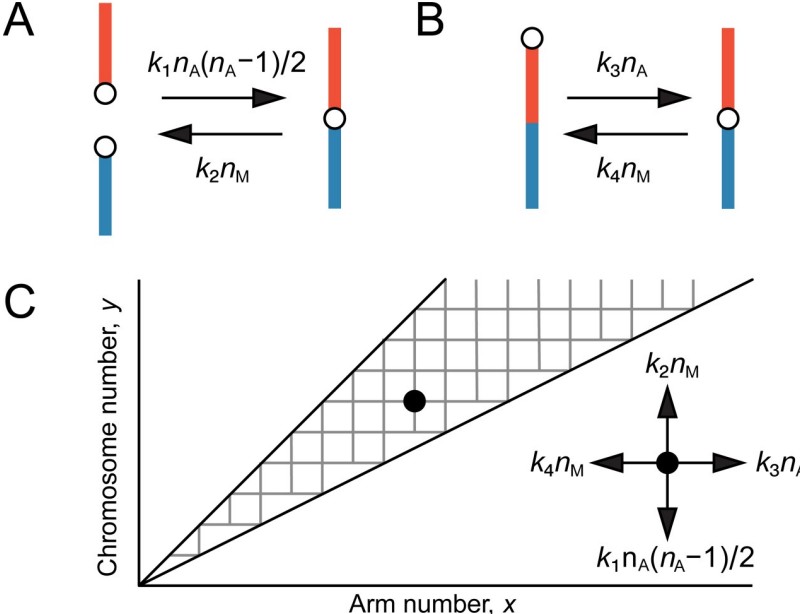

**Fig 1. Model of karyotype evolution based on the karyograph.** (A) Change in chromosome number by centric fusion or fission. The transition rates are also shown. (B) Change in arm number by centromere movement. Transition rates from acrocentric to metacentric chromosomes (A-M transition) and from metacentric to acrocentric chromosomes (M-A transition) are shown. (C) Markov process in the space of karyograph with the X-axis showing the haploid arm number (*x*) and the Y-axis showing the haploid chromosome number (*y*). $k_1$, fusion rate for a pair of acrocentric chromosomes; $k_2$, fission rate for a metacentric chromosome; $k_3$, A-M rate for an acrocentric chromosome; $k_4$, M-A rate for a metacentric chromosome; $n_A$, the number of acrocentric chromosomes; $n_M$, the number of metacentric chromosomes.

Furthermore, we applied our model to phylogenetic comparative methods using the datasets of two large groups of fish, Eurypterygii and Series Otophysi, which comprise 57% and 31% of fish species, respectively [23], for investigating differences in the trends of karyotype evolution and examining the history of karyotype evolution. Fish karyotypes are relatively conserved with 24 haploid chromosomes as the mode [24,25], although there are variations in the chromosome number among taxa. As these two monophyletic fish groups differed in the mode of chromosome number (see below), we selected these two taxa for the subsequent analysis. First, we estimated evolutionary parameters of these two groups using maximum likelihood methods based on our established model and compared the evolutionary parameters between lineages using a likelihood ratio test. Second, we reconstructed the history of the karyotype evolution of these two groups based on the estimated parameters. Third, we tested whether species with different karyotypes have different speciation and extinction rates using multi-state speciation and extinction model [26]. Finally, we made a model including polyploidization rates and applied that model to a plant group Brassicaceae.

## Results

### A probabilistic model of karyograph

To develop our model of karyotype evolution, karyotype evolution was simulated as a continuous time Markov process in the karyograph represented by a two-dimensional space of haploid arm number ($x$) and haploid chromosome number ($y$). As the arm number is not less than the chromosome number and not more than twice the chromosome number, any karyotype is within the range denoted by the following inequality (Fig 1C):

$$y \leq x \leq 2y.$$

Karyotype transitions occur via four types of chromosomal rearrangements: centric fusions, centric fissions, and two types of centromere movement (Fig 1): centromere movement from the middle to the periphery transforms metacentric chromosomes into acrocentric chromosomes and is designated as M-A transition, whereas centromere movement from the periphery to the middle transforms acrocentric chromosomes into metacentric chromosomes and is designated as A-M transition. We assumed constant probabilities of centric fusion ($k_1$) for any pair of acrocentric chromosomes and A-M transition ($k_3$) for each acrocentric chromosome. We also assumed constant probabilities of centric fission ($k_2$) and M-A transition ($k_4$) for each metacentric chromosome.

Transition rates of karyotypes can be calculated by summing the probabilities of each transition for all acrocentric and metacentric chromosomes in the karyotype. A-M transition rate ($q_{(x,y),(x+1,y)}$) is proportional to the number of the acrocentric chromosomes, $n_A$, while the transition rate of centric fission ($q_{(x,y),(x,y+1)}$) and M-A transition ($q_{(x,y),(x-1,y)}$) are proportional to the number of the metacentric chromosomes, $n_M$. The transition rate of centric fusion ($q_{(x,y),(x,y-1)}$) is proportional to the number of all combinations of acrocentric chromosomes, $n_A C_2$. Then, transition rates from a karyotype ($x$, $y$) to a neighboring karyotype ($x'$, $y'$), $q_{(x, y),(x', y')}$, can be expressed as

$$q_{(x,y),(x,y-1)} = k_{1n_A} C_2 = k_1(2y - x)(2y - x - 1)/2,$$

$$q_{(x,y),(x,y+1)} = k_2 n_M = k_2(x - y),$$

$$q_{(x,y),(x+1,y)} = k_3 n_A = k_3(2y - x),$$

$$q_{(x,y),(x-1,y)} = k_4 n_M = k_4(x - y)$$

(Fig 1C). We used the following formulas denoting the association of $n_A$ and $n_M$ with chromosome number ($y$) and arm number ($x$):

$$n_A = 2y - x,$$

$$n_M = x - y.$$

We assumed that transition to non-adjacent karyotypes is 0 (i.e., haploid chromosome and arm numbers do not change by more than one): when $x' > x + 1$, $y' > y + 1$, $x' < x - 1$ or $y' < y - 1$, transition rates $q_{(x,y),(x',y')}$ were assumed to be 0.

   Although this Markov process has infinite states, we could analytically find the stationary distribution of the karyotypes (S1 Appendix). The probability for the karyotype ($x, y$) in the stationary distribution is expressed as

$$\pi_{(x,y)} = \frac{2^y K_f^y}{(e^\Lambda - 1)K_i^x (x - y)!(2y - x)!},$$

where $\Lambda = \frac{2K_f(K_i - 1)}{K_i^2}$, $K_f = k_2/k_1$ and $K_i = k_4/k_3$. Here, $K_f$ indicates the preferential occurrence of centric fission as compared to centric fusion, while $K_i$ indicates the preferential occurrence of M-A transition as compared to A-M transition. The expectation, variance, and mode of $x$ and $y$ in the stationary distribution could be also analytically calculated (S1 Appendix).

## Performance of phylogenetic comparative analysis of karyotype evolution

To implement the phylogenetic comparative method for karyotype evolution, we built an R-script pipeline based on Mk-n or MuSSE analysis in the diversitree R package [26]. This method estimates four evolutionary parameters ($k_1$, $k_2$, $k_3$, and $k_4$) using maximum likelihood estimation. With MuSSE analysis, speciation ($\lambda$) and extinction rates ($\mu$) can also be incorporated. To evaluate the performance of the methods, we conducted the following two simulation analyses.

   First, we created a simulated dataset of karyotypes using four evolutionary parameters randomly selected on a logarithmic scale (from $10^{-4}$ to $10^{-1}$) and then estimated the parameters from the simulated dataset to test how the predicted parameters matched the parameters used to generate the dataset. For the simulation, a tree of 815 species of Eurypterygii that was used in the following analyses was used. In this simulation, we set the maximum number of chromosomes to 8 ($y_{max} = 8$) to reduce the processing time. The center of the space in the karyograph, karyotype **(6, 4)**, was set as the ancestral state. We performed 100 trials and compared the estimates with the "true" values used for the simulations. In a wide range of the parameter space, the estimates were well correlated with the "true" parameter values (Pearson's correlation coefficient, $r$ for Mk-n method = 0.83–0.91; $r$ for the MuSSE method with M0 model [i.e., no difference in speciation and extinction rates between karyotype states] = 0.81–0.95; S1 Fig). We also confirmed that the Mk-n and MuSSE methods with the M0 model gave rise to similar results ($r$ between the estimates of the two methods = 0.97–0.99). Because the processing time was much faster in the MuSSE than in the Mk-n, we used the MuSSE in the subsequent analysis regardless of whether speciation and extinction rates were variable in the model.

   Next, we tested the performance using data that was similar to the real karyotype data in fishes. The aim of this analysis was to compare among three trees with different species numbers, which correspond to three different levels of taxonomy (Clade Eurypterygii, $N$ = 815;

**Table 1. Results of maximum likelihood estimates with models with constant speciation and extinction rates.**

| Fish group | | Eurypterygii | | Otophysi | |
|---|---|---|---|---|---|
| Model | | M0 | M1 | M0 | M1 |
| No. of parameters | | 6 | 5 | 6 | 5 |
| log likelihood | | −8967 | −9187 | −5240 | -5291 |
| Estimates[1] | $k_1$ | $1.48 \times 10^{-4}$ | $8.32 \times 10^{-5}$ | $7.49 \times 10^{-4}$ | $1.22 \times 10^{-3}$ |
| | $k_2$ | $2.45 \times 10^{-3}$ | $2.29 \times 10^{-2}$ | $9.31 \times 10^{-4}$ | $4.47 \times 10^{-4}$ |
| | $k_3$ | $4.13 \times 10^{-3}$ | $6.03 \times 10^{-3}$ | $2.73 \times 10^{-2}$ | $1.51 \times 10^{-2}$ |
| | $k_4$ | $2.68 \times 10^{-2}$ | $(6.03 \times 10^{-3})$ | $1.60 \times 10^{-2}$ | $(1.51 \times 10^{-2})$ |
| | $\lambda$ | $3.83 \times 10^{-2}$ | $3.83 \times 10^{-2}$ | $5.53 \times 10^{-2}$ | $5.53 \times 10^{-2}$ |
| | $\mu$ | $1.93 \times 10^{-3}$ | $1.96 \times 10^{-3}$ | $2.28 \times 10^{-7}$ | $1.47 \times 10^{-6}$ |
| | $K_f$ | 16.5 | 275.2 | 1.2 | 0.4 |
| | $K_i$ | 6.5 | (1) | 0.6 | (1) |

The results with $y_{max} = 35$ are shown here.

[1]$k_1$, fusion rate coefficient; $k_2$, fission rate coefficient; $k_3$, A-M transition rate coefficient; $k_4$, M-A transition rate coefficient; $\lambda$, speciation rate; $\mu$, extinction rate. $K_f = k_2/k_1$, fission/fusion bias; $K_i = k_4/k_3$, M-A/A-M transition bias. Parentheses indicate constrained parameters used for the null model ($K_i = 1$).

Order Cyprinodontiformes, $N = 80$; Family Goodeidae, $N = 29$). In this simulation, we used the best parameters estimated in the following analysis of Eurypterygii (M0 model, Table 1) as the "true" values and **(24, 24)** as the "true" ancestral state of the most recent common ancestor (MRCA). We repeated the simulation and subsequent estimation 100 times. We found little deviation in the estimates from the "true" values using the tree of Class Eurypterygii with 815 species (mean absolute value of $\log_{10}$ estimate/"true" value of $k_1 = 0.065$, $k_2 = 0.11$, $k_3 = 0.040$, and $k_4 = 0.036$). The deviation was much higher when smaller numbers of species were used (Order Cyprinodontiformes, mean absolute value of $\log_{10}$ estimate/"true" value of $k_1 = 0.21$, $k_2 = 0.61$, $k_3 = 0.14$, and $k_4 = 0.12$; Family Godeidae, mean absolute value of $\log_{10}$ estimate/"true" value of $k_1 = 0.62$, that of $k_2 = 2.3$, that of $k_3 = 0.68$, that of $k_4 = 1.1$; S2 Fig). Among the four parameters, $k_2$ had the highest error rate in all the cases.

Furthermore, ancestral state reconstruction at the MRCA was conducted in 100 simulations for each tree. Analysis using the Eurypterygii tree with the age of MRCA = 156.3 million years ago showed that the range of the estimated ancestral chromosome number was narrow (average 95% range = 20.7–24.6; see Materials and Methods), whereas that of the estimated ancestral arm number was broad (average 95% range = 23.9–41.8, S3 and S4 Figs). This uncertainty of the ancestral arm number was greatly reduced when the MRCA was younger (Cyprinodontiformes with the age of MRCA = 66.7 million years ago, average 95% range of arm number = 23.8–32.6; Godeidae with the age of MRCA = 19.4 million years ago, average 95% range of arm number = 24.0–25.4).

## Fish karyograph

We chose fish species to apply our model for the analysis of empirical data. Karyotypes of 2,587 species of teleost fish were first plotted on the karyograph (S5 and S6 Figs). The mode was at the karyotype **(24, 24)**: here, the first letter indicated the arm number and the latter indicated the chromosome number. This mode represents a karyotype with 24 acrocentric chromosomes and no metacentric chromosomes. Two local maxima were also observed at the karyotypes **(47, 25)** and **(54, 27)**, which are similar to the karyotype **(24, 24)** in chromosome number, but largely differ in arm number. In these two local maxima, the majority of chromosomes are metacentric. Reflecting this, a bimodal frequency distribution of the acrocentric chromosomes was observed when plotted

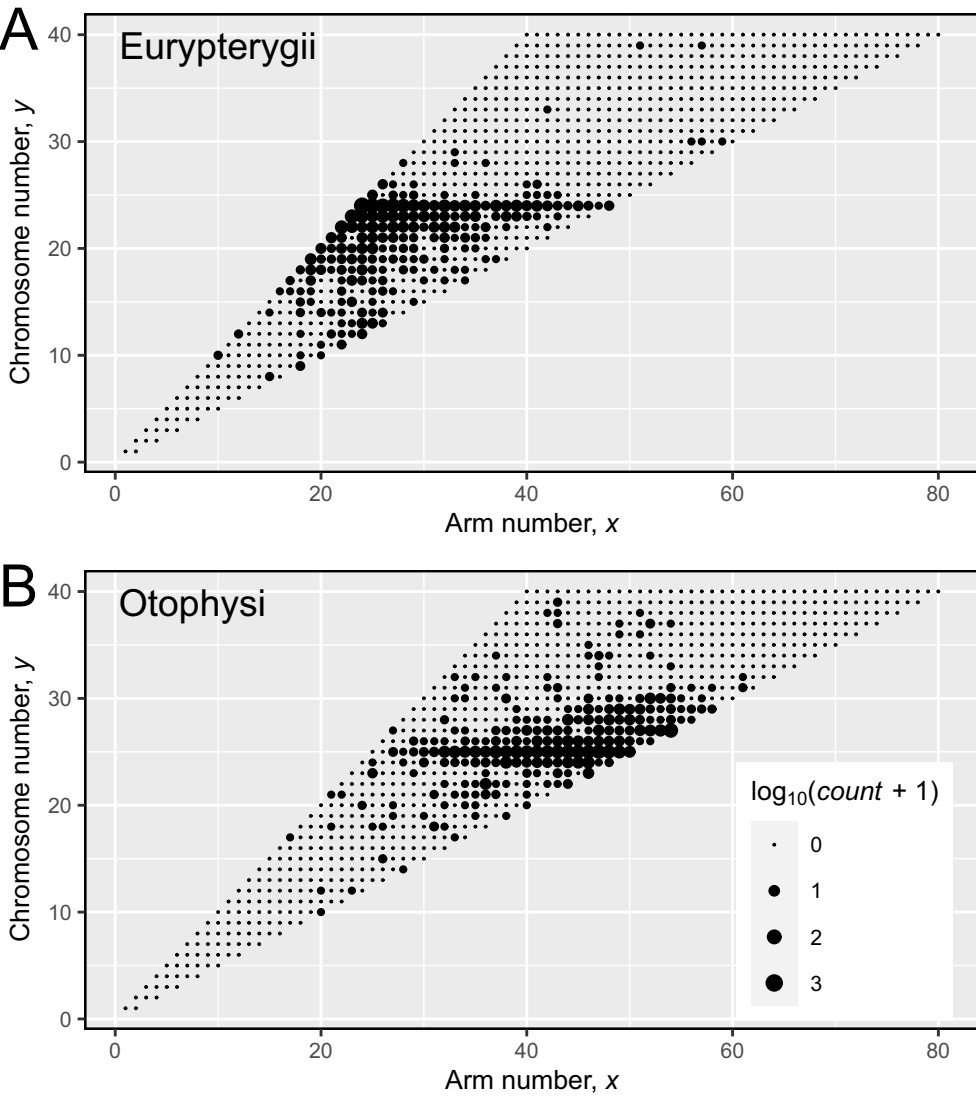

**Fig 2. Karyograph of Eurypterygii and Otophysi.** Karyotypes of 1360 species of Eurypterygii (A) and 1026 species of Otophysi (B) were plotted on the karyograph. The sizes of the circle indicate $\log_{10}$ of the species number plus one. Eight outlier species of Eurypterygii and four outlier species of Otophysi are not shown here (see the text).

for all teleosts (S7A Fig). Difference between the karyotype **(24, 24)** and two local maxima reflected phylogenetic groups (Fig 2). Most fishes belonging to the monophyletic Eurypterygii ($N = 1{,}368$) group exhibited karyotype **(24, 24)** or similar karyotypes (Figs 2A and S8A), whereas most fishes belonging to other large monophyletic group, series Otophysi ($N = 1{,}030$), exhibited karyotypes **(47, 25)**, **(54, 27)**, or similar karyotypes (Figs 2B and S8B). Hence, the two groups differed in the frequency distribution of the acrocentric chromosomes (S7B and S7C Fig). Within each group, the karyotype was relatively conserved even among different orders (S1 Table).

## Phylogenetic comparative analysis with constant speciation and extinction rates

We estimated four evolutionary parameters ($k_1$, fusion rate coefficient; $k_2$, fission rate coefficient; $k_3$, A-M transition rate coefficient; $k_4$, M-A transition rate coefficient) for Eurypterygii

($N$ = 815) and Otophysi ($N$ = 503) using the MuSSE method with constant speciation and extinction rates among karyotypes (M0 model). Here, we set the maximum limit of the chromosome number ($y_{max}$) to 35 (i.e., diploid chromosome number = 70): for validation of the use of $y_{max}$ = 35, see the Materials and Methods and S1 Appendix. Table 1 shows the estimated parameters. Eurypterygii demonstrated a high M-A/A-M transition bias ($K_i$ = 6.5), whereas Otophysi demonstrated a low M-A/A-M transition bias ($K_i$ = 0.6). We calculated the likelihood of the model with $K_i$ = 1 (M1 model) and found that the M-A/A-M transition bias was significantly higher than 1 in Eurypterygii and lower than 1 in Otophysi (likelihood ratio test, $P < 2.2 \times 10^{-16}$ in both). This suggests that the differences in the bias of intra-chromosomal centromere movement are likely to be responsible for different karyotype distributions between these two groups.

Next, we reconstructed ancestral states at each node using maximum likelihood estimation (Figs 3A and 3B, and 4A and 4B). In both Eurypterygii and Otophysi, the ancestral state reconstruction of the MRCA had a narrow range for the estimated chromosome number (Eurypterygii, 95% range = 23–27; Otophysi, 95% range = 25–32) but a broad range for the estimated arm number (Eurypterygii, 95% range = 26–46; Otophysi, 95% range = 30–51; S9 and S10 Figs). The ancestral state reconstruction of the MRCA of Order Cyprinodontiformes and Family Goodeidae, which are nested within Eurypterygii, have a narrower range of arm numbers (Cyprinodontiformes, 95% range of arm number = 23–27; Goodeidae, 95% range of arm number = 24–25; S9 Fig). These results are in concordance with the simulation analysis, which also showed that the range of the reconstructed state of the MRCA became narrower when the MRCA was younger (S3 and S4 Figs).

## Phylogenetic comparative analysis with variable speciation and extinction rates among karyotypes

One of the unique characteristics of the fish karyotype is the predominance of species at the distribution mode. In Eutypterygii, 29% (233/815) of the species has the same karyotype as **(24, 24)**. We simulated karyotype evolution using the estimated parameters, the reconstructed ancestral state, and the Eurypterygii tree (10,000 times) to investigate how many percentages of species have the karyotype **(24, 24)**. The mean percentage of species located at **(24, 24)** was only 0.28% (S11A Fig). None of the 10,000 simulations exceeded more than 2.7%. Furthermore, the peak of the distribution was distant from **(24, 24)** (the mode of mean proportion = **(26, 23)**, S11B Fig).

High speciation rates and/or low extinction rates may be able to account for the prevalence of certain traits. To investigate this possibility, we tested whether species with the karyotype **(24, 24)** have a different speciation and extinction rate than species with other karyotypes in Eutypterygii by incorporating the different speciation and extinction rates among karyotypic states into our model (M2 model). The parameter estimation suggested that species with the karyotype **(24, 24)** had a slightly higher speciation rate and a much lower extinction rate than those with the other karyotypes (speciation rate of **(24, 24)**, $\lambda_m = 1.56 \times 10^{-1}$, speciation rate of the other karyotypes, $\lambda_{other} = 1.25 \times 10^{-1}$, extinction rate of **(24, 24)**, $\mu_m = 2.63 \times 10^{-7}$; extinction rate of the other karyotypes, $\mu_{other} = 1.56 \times 10^{-1}$; Table 2). A particularly striking difference was observed in the extinction rates, which differed by over five orders of magnitude between the karyotypes. Karyotypes other than **(24, 24)** had even higher extinction rates than their speciation rates, indicating that these karyotypes would likely go extinct. Comparison with the model with constant speciation and extinction rates (M0' model) showed that the model with variable speciation and extinction rates was more highly supported (likelihood ratio test, $p$-value $< 2.2 \times 10^{-16}$; S2 Table).

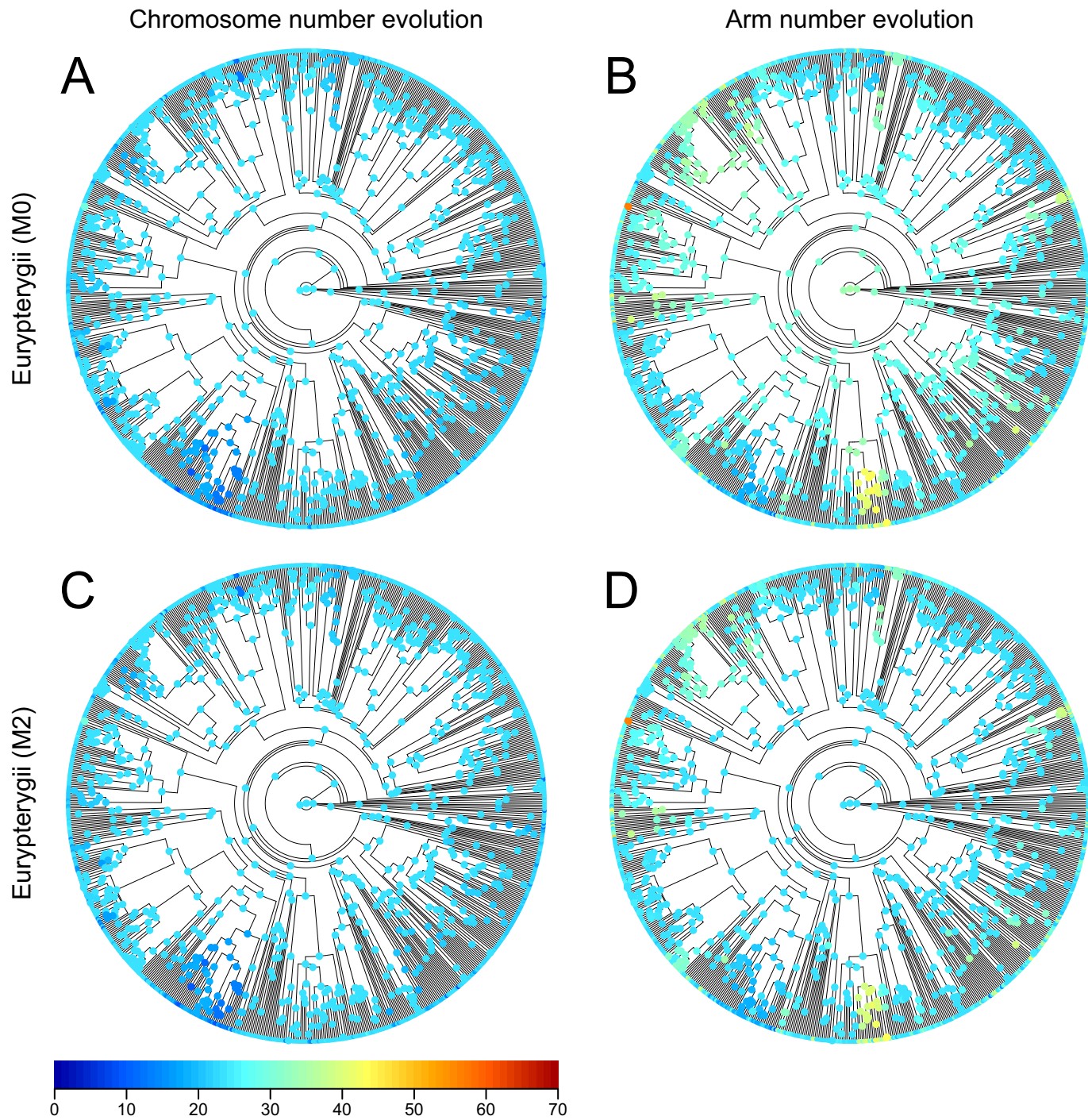

**Fig 3. Phylogenetic tree of Eurypterygii with the ancestral state reconstruction of karyotypes.** The circles at the nodes indicate heat maps of mean values of chromosome numbers (A, C) or arm numbers (B, D) in the marginal ancestral reconstruction of karyotypes of Eurypterygii with M0 model (A, B) and M2 model (C, D). The colored points on the tips indicate the karyotype of extant species.

As three species of Eurypterygii were filtered out in our analysis due to the large chromosome number ($y > 35$) or polyploidy, we examined whether this filtering biased our results. A subgroup of Eurypterygii, series Eupercaria, included no species that were filtered out. The results of the analysis using only Eurpercaria showed that the extinction rate was much lower

## Chromosome number evolution

## Arm number evolution

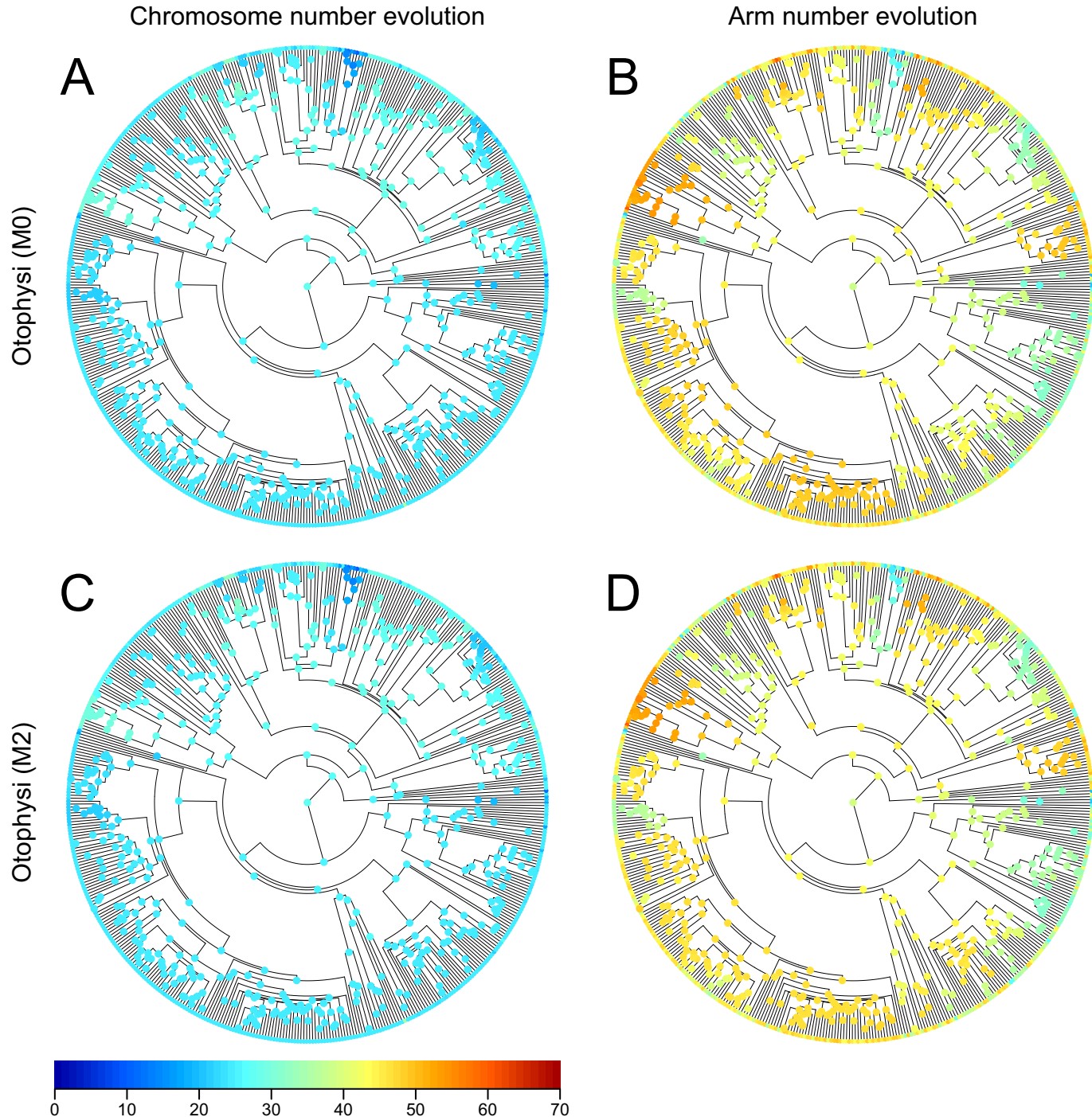

**Fig 4. Phylogenetic tree of Otophysi with the ancestral state reconstruction of karyotypes.** The circles at the nodes indicate heat maps of mean values of chromosome numbers (A, C) or arm numbers (B, D) in the marginal ancestral reconstruction of karyotypes of Otophysi with M0 model (A, B) and M2 model (C, D). The colored points on the tips indicate the karyotype of extant species.

in the karyotype **(24, 24)**, although the speciation rate was also slightly lower in the karyotype **(24, 24)** than in other karyotypes (S2 Table). The model with variable speciation and extinction rates was again more highly supported than the model with constant speciation and extinction rates (likelihood ratio test, $p$-value $< 2.2 \times 10^{-16}$, S2 Table).

**Table 2. Results of maximum likelihood estimates with models with variable speciation and extinction rates.**

| Fish group | | Eurypterygii | | Otophysi | |
|---|---|---|---|---|---|
| Model | | M2 | M3 | M2 | M3 |
| No. of parameters | | 8 | 7 | 8 | 7 |
| log likelihood | | −8727 | −9086 | −5230 | -5293 |
| Estimates[1] | $k_1$ | $1.73\times10^{-4}$ | $1.03\times10^{-4}$ | $7.78\times10^{-4}$ | $1.19\times10^{-3}$ |
| | $k_2$ | $3.36\times10^{-3}$ | $2.10\times10^{-2}$ | $1.04\times10^{-3}$ | $4.69\times10^{-4}$ |
| | $k_3$ | $5.41\times10^{-3}$ | $5.85\times10^{-3}$ | $3.07\times10^{-2}$ | $1.51\times10^{-2}$ |
| | $k_4$ | $1.94\times10^{-2}$ | $(5.85\times10^{-3})$ | $1.75\times10^{-2}$ | $(1.51\times10^{-2})$ |
| | $\lambda_m$ | $1.56\times10^{-1}$ | $1.01\times10^{-1}$ | $2.25\times10^{-1}$ | $1.53\times10^{-1}$ |
| | $\mu_m$ | $2.63\times10^{-7}$ | $3.86\times10^{-6}$ | $6.06\times10^{-9}$ | $7.34\times10^{-6}$ |
| | $\lambda_{other}$ | $1.25\times10^{-1}$ | $6.48\times10^{-2}$ | $5.20\times10^{-2}$ | $5.62\times10^{-2}$ |
| | $\mu_{other}$ | $1.56\times10^{-1}$ | $4.65\times10^{-2}$ | $1.21\times10^{-2}$ | $1.09\times10^{-2}$ |
| | $K_f$ | 19.4 | 203.9 | 1.3 | 0.4 |
| | $K_i$ | 3.6 | (1) | 0.6 | (1) |

The results with $y_{max} = 35$ are shown here.

[1]$k_1$, fusion rate coefficient; $k_2$, fission rate coefficient; $k_3$, A-M transition rate coefficient; $k_4$, M-A transition rate coefficient; $\lambda_m$, speciation rate of the karyotype at the local maxima; $\mu_m$, extinction rate of the karyotype at the local maxima; $\lambda_{other}$, speciation rate of the other karyotypes; $\mu_{other}$, extinction rate of the other karyotypes. The local maximum of Eurypterygii is only **(24, 24)**, while the local maxima of Otophysi are **(47,25)** and **(54,27)**. $K_f = k_2/k_1$, fission/fusion bias; $K_i = k_4/k_3$, M-A/A-M transition bias. Parentheses indicate constrained parameters used for the null hypotheses ($K_i = 1$).

We also applied the model with variable speciation and extinction rates to Otophysi: we allowed the speciation and extinction rates at the two local maxima of the karyotypes, **(47,25)** and **(54,27)**, to differ from those of the other karyotypes. Species at the two local maxima were estimated to have a much lower extinction rate and a slightly higher speciation rate than species with other karyotypes (Table 2). This model with variable speciation and extinction rates was more statistically supported than the model with constant speciation and extinction rates (likelihood ratio test, $p$-value = $3.7\times10^{-11}$; S2 Table).

It should be noted that in the model with variable speciation and extinction rates (M2 model), the directions of fission/fusion bias or M-A/A-M transitions were qualitatively similar to those estimated in the model with constant speciation and extinction rates (M0 model). Furthermore, the likelihood ratio test comparing the M2 model with a model with a constraint of $K_i = 1$ and variable rates of speciation and extinction (M3 model) suggested that $K_i$ is higher than 1 in Euryptergii and lower than 1 in Otophysi (likelihood ratio test, $p$-value $< 2.2\times10^{-16}$ in both; Table 2).

Next, we reconstructed the ancestral states of nodes using the models with variable speciation and extinction rates (M2 model) (Figs 3C and 3D, and 4C and 4D). The ancestral state of Eurypterygii was estimated to be **(24, 24)** (Fig 3C and 3D) with very small error rates (MRCA of Eurypterygii, 95% ranges of both chromosome and arm numbers were within 24; S12 Fig). In contrast, the ancestral state reconstruction in Otophysi still has large uncertainty (MRCA of Otophysi, 95% range of chromosome number = 25–31, that of arm number = 30–52; S13 Fig). Reconstructed evolutionary trajectories were overall similar between the M0 and M2 models (Fig 4).

## Characterization of tree branches

To characterize evolutionary rates on tree branches, we determined the branch length fitted to the reconstructed karyotype transitions using a maximum likelihood method with a likelihood

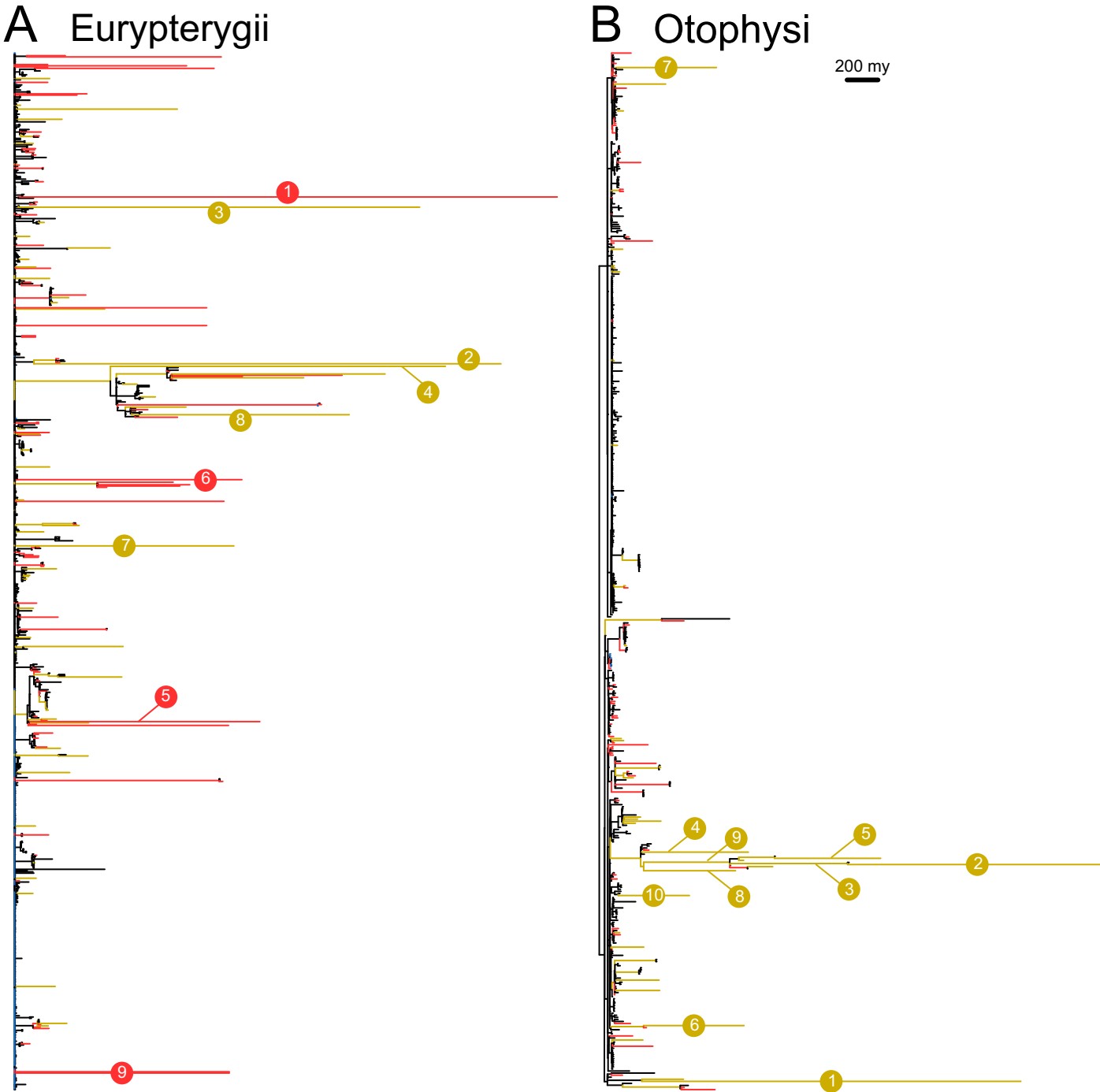

**Fig 5. Phylogenetic tree with fitted branches of Eurypterygii (A) and Otophysi (B).** The branch length represents the relative difference of karyotypes between nodes calculated by maximum likelihood estimation with the estimated parameters (see Materials and Methods). The color of the branches indicates the classification of branches (black, "Expected"; blue, "Conservative"; yellow, "Rapid"; red, "Unusual"). The top 10 longest branches were numbered. The numbers on the branches correspond to the branches in Table 2. The scale bar indicates 200 million years (my).

function composed of the estimated evolutionary parameters with the M2 model; the branch length reflected the relative difference in karyotypes between nodes. The fitted branch lengths are shown in Fig 5. To classify the branches, we categorized them into four categories:

"Expected," "Conservative," "Rapid," and "Unusual" (see Materials and Methods and S14 Fig). When transitions were within the range expected from the actual branch length in the ultra-metric phylogenetic tree, the transitions were categorized as "Expected." When transitions were out of the range expected from the actual branch length but within the range expected from the fitted branch length calculated using the estimated parameters, the transitions were categorized as either "Conservative" or "Rapid." In cases where the fitted branch length was longer than the actual branch length, the transitions were categorized as "Rapid;" otherwise, they were categorized as "Conservative." When transitions were out of the range expected from both the actual and fitted branch lengths, the transitions were categorized as "Unusual."

A total of 89% and 82% of branches were categorized into the "Expected" category for Eurypterygii and Otophysi, respectively (S3 Table). "Rapid" and "Unusual" categories were significantly enriched in terminal nodes in Eurypterygii (S3 Table; Fisher's exact test: $p$-value = $8.10 \times 10^{-6}$ in "Rapid," $p$-value = $8.97 \times 10^{-14}$ in "Unusual") and in Otophysi (Fisher's exact test: $p$-value = $2.07 \times 10^{-2}$ in "Rapid," $p$-value = $1.63 \times 10^{-8}$ in "Unusual"). The top ten longest branches of each group are listed in S4 Table. The Eurypterygii tree had long terminal branches in different groups (red branches in Fig 5A), which contrasted with Otophysi with long branches relatively clustered in the internal and terminal branches of the genus *Liobagrus* (Fig 5B).

## Inclusion of polyploidization in the model

Polyploidization is another path of karyotype evolution that was not assumed in our model above. In particular, plant species have undergone frequent polyploidization [1]. We incorporated a constant polyploidization rate ($k_5$) into our model (M4 model) and applied it to the *Brassicaceae* plant species, where both arm and chromosome numbers are available for 39 species (S5 Table and S15 Fig). Although we could not find maximum likelihood with the model without polyploidization ($k_5 = 0$) because of excessive processing time, we could find maximum likelihood in the model with polyploidization (M4 model) with $k_5 = 9.31 \times 10^{-3}$ (/million years; S6 Table). Using this model, we constructed ancestral states at each node and found seven possible events of polyploidization (Figs 6 and S16). Although the ancestral state reconstruction at the oldest node (i.e., MRCA) indicated a broad range of the inferred chromosome number (95% range of chromosome number = 8–16; S17 Fig), that of the second oldest node indicated a narrow range of the inferred chromosome number around 8 (95% range of chromosome number = 7–10; S17 Fig). We also performed the same analysis using the dataset where species reported to be polyploid were removed, and 29 species remained. The analysis detected two additional cases of polyploidization that have not been reported in the literature (S16C and S16D Fig). Both evolutionary parameters (i. e., $K_f$ and $K_i$) and reconstructed ancestral states were similar to the former estimates using all 39 species (S6 Table and S16 Fig).

## Discussion

We established a probabilistic model of karyotype evolution involving both chromosome and arm numbers, evaluated their performance using simulated data, and applied our model to two large groups of fish and one small group of plant to understand the evolutionary trajectories of karyotype evolution in these taxa. The majority of previous studies on karyotype evolution considered only chromosome number and ignored the chromosome morphology [8,9,11–13,27,28]. These models would be enough for taxa whose chromosomes are holocentric [13] or for taxa in which the centromere movement is rare [29]. In monocentric taxa, however, the chromosome number evolution is tightly linked to the evolution of chromosome morphology. Without centromere movement, haploid chromosome number, $y$, would be

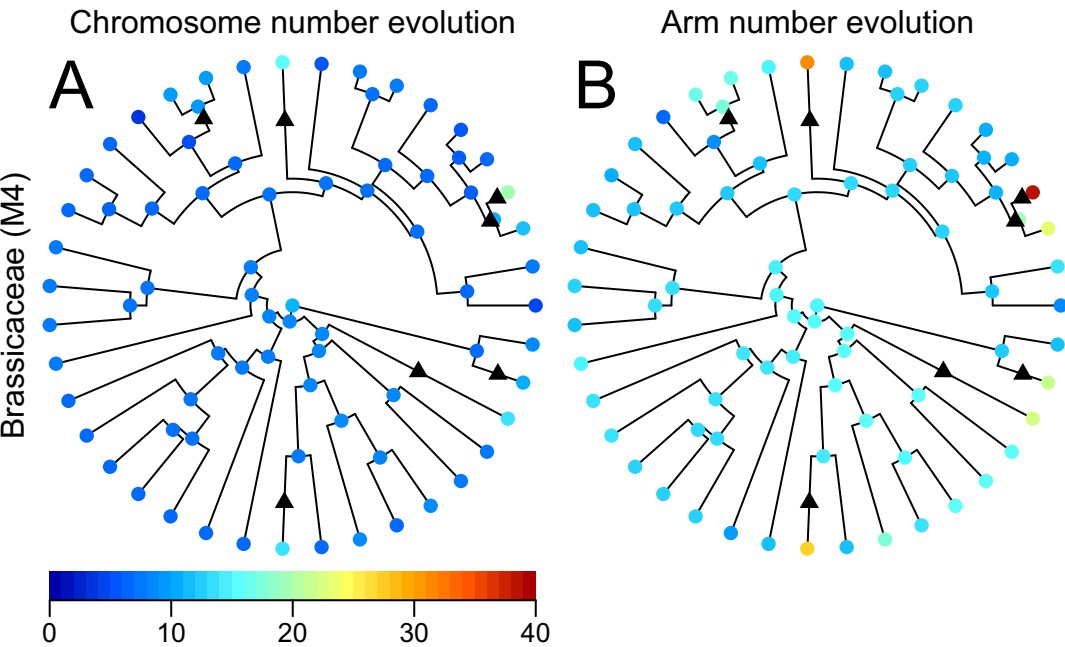

**Fig 6. Phylogenetic tree of Brassicaceae with the ancestral state reconstruction of karyotypes.** The circles at the nodes indicate heat maps of mean values of chromosome numbers (A) and arm numbers (B) in the marginal ancestral reconstruction of karyotypes inferred with the M4 model. The colored points on the tips indicate the karyotype of extant species. Black triangles show possible polyploidization events.

limited within the range of $x/2 \leq y \leq x$; here, $x$ is the ancestral haploid arm number. This model would be unrealistic for many taxa [18]. Additionally, the rate of centric fusion/fission would depend on the number of source chromosomes, acrocentric chromosomes for fusion and metacentric chromosomes for fission, indicating that the evolution of chromosome and arm numbers is not independent. Furthermore, our estimation of parameters indicates that the rates of centromere movement are even higher than the rates of fusion/fission for several taxa (Table 1). A recent study reported a two-state model of chromosome morphology evolution, which assumes that karyotype morphology is either a "matched" state with the proportion of acrocentric chromosomes $p_A = 0\%$ or $100\%$ or an "unmatched" state with $0\% < p_A < 100\%$ [30]. This previous model also assumes that the transition of chromosome morphology is independent of chromosome number. As our model does not make these assumptions, our model may be able to evaluate the actual karyotype evolution better.

Comparison of whole genome sequences of multiple taxa is another approach for inferring past karyotype evolution. Such synteny-based analysis proposed that the ancestral karyotype of teleosts may be similar to that of *Oryzias latipes* [31,32], which has a chromosome number of 24. The chromosome number of the MRCA karyotype inferred in the M2 Model was also 24, suggesting that our estimate of ancestral state reconstruction is consistent with that inferred in the synteny-based inference. In Brassicaceae, synteny analyses suggested that the ancestral chromosome number is 8 [33], and species with smaller numbers of chromosomes, including *Arabidopsis thaliana* (haploid chromosome number = 5), are derived from chromosomal fusions [34,35]. Our analysis also indicates a trend of reduction in chromosome number, which is consistent with this prediction. Synteny analysis of whole genome sequence data can even provide information about the evolution of gene order at a fine scale, whereas the position of the centromere is still difficult to identify using only the next generation sequence data.

Once complete genome assembly of diverse taxa becomes feasible and the positions of centromeres can be easily mapped to the genome sequences, our model could be integrated with the synteny-based analysis to make the inference of karyotype evolution more precise.

Our method still has several limitations. First, the number of species affects the precision of the estimation. Our simulation analysis showed that the estimation was quite precise with 815 species, whereas the uncertainty increased with lower numbers of species. Even if one is interested only in a small taxon group, a wider sampling of taxa would be better for this method. Second, we assumed that the karyograph space is within a certain range because of the computational limit. However, theoretically, a karyotype can move in infinite space. Although our validation analyses justified the use of karyograph space limit in the present study (see Materials and Methods and S1 Appendix), any user interested in other taxa need to evaluate their specific chromosome and arm number limits. The limit of karyograph space will matter particularly in taxa with high polyploidization rates, because polyploidization can multiply both chromosome and arm numbers and may exceed the limit set by the user. In the present study on the teleosts, we excluded polyploid species before analysis, because it requires excessive computational time for models including polyploidization rates and with a higher maximum number of chromosomes ($y_{max}$). There is room for improvement in the processing time with the use of programming systems other than the R language. Third, in the present study, we assumed that the parameters were constant across the phylogenetic tree analyzed. However, some taxa may change the parameters very rapidly. For example, mammals shift the direction of female meiotic drive frequently between the drives favoring fusion and fission [21,36], suggesting that the application of our model to any large mammalian group with constant parameters is not recommended. Nevertheless, our model would be applicable for a comparison between small groups of mammals. If the factors determining the direction of the female meiotic drive are demonstrated, it would be possible to include such factors in our model. Finally, we assumed that the change in chromosome number occurs via centric fusion or fission. However, chromosome number can change by non-centric mechanisms, such as telomere fusion and non-centric fission. Telomere fusion can generate a dicentric chromosome, which can be deleterious [37]. As non-centric fission splits one chromosome into two, with only one having a centromere and the other lacking a centromere, it can have deleterious effects [38]. Therefore, we did not consider these types of fusion and fission. When some taxa, however, have higher rates of this type of karyotype evolution, these rates should also be included as parameters.

We demonstrated that the patterns of karyotype evolution can differ between taxonomic groups. M-A transition rate ($k_4$) was higher than the A-M transition rate ($k_3$) in Eurypterygii and *vice versa* in Otophysi (Tables 1 and 2). As the transition rates are the products of mutation rates and fixation probabilities, any factors that can bias these two would explain the taxonomic differences. Pericentric inversion is one of the mechanisms of centromere movement [18,39,40]. The presence of preferential mutation direction of pericentric inversion was suggested previously [18] but still lacks empirical evidence. The bias in fixation probability via non-random transmission of different karyotypes in heterokaryotypes would be another possible mechanism leading to the bias of centromere movement [21]. Novitski proposed a hypothesis that recombination within the pericentric inversion results in the production of differently sized chromosomes with different centromere positions, and female meiotic drive for chromosome size, if any, can lead to the segregation distortion of sister chromatids exhibiting centromeres at different positions [41]. Supporting this idea, non-mendelian segregation in heterozygotes of chromosomes with differently positioned centromeres was observed in flowering plant *Rumex acetosa* [42], onion fly *Hylemya antiqua* [43], and blowfly *Lucilia cupina* [44], although there are cases without any bias of segregation [45].

We found that karyotype is associated with extinction rates. For example, in Eurypterygii, a higher extinction rate and a lower speciation rate were observed in species that exhibit karyotypes other than **(24, 24)** (Table 2). This observation seems to contradict previous studies demonstrating that speciation is often associated with chromosome evolution in reptiles and mammals [9,14,15]. Our present study demonstrated that the roles of karyotype in speciation and extinction may differ between taxa. Currently, we are unsure about the cause of this effect of karyotypes on extinction rates. One possible cause would be molecular and cellular constraints associated with mitosis and meiosis. As far as the genome size remains constant, a negative correlation would exist between the chromosome number and the size of each chromosome. Long chromosomes, such as those with the arms longer than half of the spindle axis, may result in improper segregation [46]. Chromosomes that are smaller than a particular limit also tend to segregate improperly during meiosis [47]. Therefore, an optimal karyotype may exist for proper segregation in Eurypterygii, and constraints for proper segregation may differ between taxa. Further studies on some species with karyotypes other than the optimal ones that have survived until today (S4 Table) may provide insights into the mechanisms. Finally, although we found an association between karyotype and speciation/extinction rates in multiple taxa, it should be noted that binary-state speciation and extinction models can produce false positive results [48,49], so our results should be interpreted with caution.

In conclusion, the inclusion of chromosome arm number in the karyotype evolution model can improve our understanding of the tempo and mode in chromosome evolution and possible roles of karyotype in speciation and extinction. Our model-based inference has the potential to become complementary to synteny-based approaches for the inference of karyotype evolution.

## Materials and methods

### Fish karyotype data

We compiled data on fish karyotypes of teleost fishes from the literature [50]. We used the fundamental numbers (NF1) as arm numbers (*AN*). The complied data contained 2,716 species. Fishes reported as polyploid (116 species), having odd numbers of chromosomes (9 species), karyotyped only during meiosis (4 species) were excluded. We used female karyotypes when sex differences were observed in the karyotypes. B chromosomes, if any, were not counted. When different karyotypes were reported from a single species (i.e., inter-population variations), we selected a karyotype using the criteria described below. For the chromosome number, one was randomly chosen. For the arm number, we chose the median when odd-numbered cases were observed and chose one randomly from either of the two nearest to the medians when even-numbered cases were observed. In total, 2,587 species were used for plotting the fish karyotype (S1 Table). We used taxonomic names in accordance with the commonly accepted classification system [51], following Arai (2011) [50].

To apply our probabilistic model to empirical data, we used two large monophyletic groups, Eurypterygii and Otophysi; monophyly of these groups was reported in previous phylogenetic studies [52,53]. Teleost fishes are composed of four subdivisions, two basal small subdivisions, Osteoglossomorpha and Elopomorpha, and two large advanced subdivisions, Otocephala and Euteleostei [51]. Most orders in Euteleostei exhibit similar karyotype distributions [25] with karyotype **(24, 24)** as the mode (S1 Table), except for Salmoniformes, which experienced whole-genome duplication in its ancestor and possesses largely different karyotypes (S1 Table). Therefore, we inferred that Euteleostei, except Salmoniformes, have similar evolutionary parameters and determined to use a monophyletic group excluding Salmoniformes and related sister orders (Argentiniformes, Osmeriformes, Esociformes and Stomiiformes) [52].

Because no karyotype data was available for Ateleopodiformes, we used Eurypterygii, which is a monophyletic group composed of the rest of the orders in Euteleostei. Series Otophysi, a sub-group of Otocephala, includes orders having karyotypes with high arm numbers, which is largely different from Eurypterygii (Fig 2 and S1 Table). Because Clupeiformes, a basal order of Otocephala, exhibits karyotypes similar to Euteleostei (S1 Table), we inferred that the evolutionary parameters changed in the ancestors of Otophysi after divergence from Clupeiformes. Only two karyotyped species were available from an outgroup of Otophysi, Gonorynchi-formes. Therefore, we decided to use Otophysi for comparison with Eurypterygii. Eurypterygii and Otophysi include 60% and 32% of teleost species, respectively, and 57% and 31% of fish species, respectively [23]. Karyotype data used are available from Dryad (https://datadryad. org/stash/dataset/doi:10.5061/dryad.s4mw6m966) [54].

## Maximum likelihood estimation of evolutionary parameters

For the maximum likelihood estimation of evolutionary parameters, we used the Mk-n model and MuSSE model with a phylogenetic tree and discrete trait data [55] implemented in an R package, diversitree (https://cran.r-project.org/web/packages/diversitree/) [26]. Karyotypes were assumed to evolve within a definite space with an arbitrarily assigned maximum number of chromosomes, $y_{max}$ (see S1 Appendix). Each karyotype was given a unique number and treated as different states (e.g. 665 states when $y_{max} = 35$). The likelihood function was prepared by make.mkn and make.musse function in the diversitree package for the Mk-n and the MuSSE analyses, respectively. The constraints of transition rates were designed based on the transition rates of our probabilistic model. As the constraint function in the diversitree package is not applicable for large transition matrixes, we wrote a custom script. Maximum likelihoods were estimated using the find.mle function in the diversitree package. All priors were set to 0.1 for the Mk-n whereas the starting.point.musse function was used to set priors for the MuSSE analysis.

## Simulation analyses

To evaluate the performance of the parameter estimation, we performed two simulation analyses. First, we simulated karyotype evolution with randomly selected parameters and then estimated the parameters using the Mk-n and MuSSE methods (M0 model). The maximum limit of chromosome number ($y_{max}$) was set to eight to reduce the processing time. The ancestral state was set to the center of the space, **(4, 6)**. The empirical phylogenetic tree of 815 species of Eurypterygii used for data analysis (see below) was also used in the simulation. In each trial, we randomly selected logarithm of $k_1$, $k_2$, $k_3$, and $k_4$ to base 10 from −4 to −1 and used them for making the transition matrix ($Q$), followed by simulating karyotype evolution using the sim.character function with the "mkn" model. Using the dataset generated by the simulations, we estimated four parameters using the Mk-n and MuSSE methods (M0 model). One hundred trials were performed.

Second, we conducted simulations using four parameters, $k_1$, $k_2$, $k_3$, and $k_4$, inferred from the MuSSE analysis (M0 model) of Eurypterygii with $y_{max} = 35$ (Table 1). The ancestral state was set as **(24, 24)**, which is the mode of the distribution of karyotypes of Eurypterygii. To investigate how the number of species influences the inference of the parameters and the ancestral state, we selected three clades: Clade Eurypterygii; Order Cyprinodontiformes, which is within Eurypterygii; Family Goodeidei, which is within Cyprinodontiformes. In each trial of each group, we generated a karyotype dataset using the sim.character function and estimated four parameters using the simulated data with the MuSSE method (M0 model). We also conducted marginal ancestral state reconstruction (ASR) of the most recent common ancestor

(MRCA) of each clade using the asr.marginal function. For ASR, we used the parameters inferred from the true (not simulated) dataset (Table 1).

To test the enrichment of species with the karyotype **(24, 24)** in Eurypterygii, we simulated karyotype evolution using the Eurypterygii tree with the ancestral karyotype set to **(28, 26)**, which is the mode of the ancestral state reconstruction of MRCA of Eurypterygii using the MuSSE method (M0 model).

## Phylogenetic comparative analysis using fish phylogeny

To apply our phylogenetic comparative method to fish phylogeny, we downloaded an ultra-metric phylogeny of 7,822 fish species, including all teleost orders, from Dryad (https://doi. org/10.5061/dryad.j4802) [56]. We used species or subspecies with identical names between the phylogenetic tree and the karyotype data. Maximum likelihood estimation of the parameters was conducted for two taxonomic groups, Eurypterygii and series Otophysi. We first estimated the parameters with two values of $y_{\max}$ (35 and 40) in the M0 model (see below). More than 96% of species in each taxonomic group were included in the defined spaces (815/817 and 503/517 species of Eurypterygii and Otophysi, respectively, when $y_{max}$ = 35). We confirmed that the results with $y_{\max}$ = 35 and $y_{\max}$ = 40 were not substantially different (S7 Table), indicating that the choice of $y_{\max}$ did not substantially affect the results. Therefore, we used $y_{\max}$ = 35 for subsequent analysis, which required much less computational time.

We conducted MuSSE analysis with four different models, M0-M3 models for the analysis of fish phylogeny. The M0 model is a model with six free parameters including four evolutionary parameters ($k_1$, $k_2$, $k_3$, and $k_4$) and speciation ($\lambda$) and extinction rates ($\mu$), which are constant between karyotypes. The M1 model is a model with five free parameters including the three parameters, $k_1$, $k_2$, and $k_3$, with a fixed constraint $K_i = k_4/k_3 = 1$ and constant speciation ($\lambda$) and extinction ($\mu$) rates. The M2 model is a model with eight free parameters including the four parameters, $k_1$, $k_2$, $k_3$, and $k_4$, and two different speciation and extinction rates between two different karyotype groups, the local maxima of karyotypes (karyotype **(24, 24)** in Eurypterygii and karyotypes **(47,25)** and **(54,27)** in Otophysi) ($\lambda_m$ and $\mu_m$) rate and other karyotypes ($\lambda_{\text{other}}$ and $\mu_{\text{other}}$). The M3 model is a model with seven free parameters including three parameters, $k_1$, $k_2$, and $k_3$, with a fixed constraint $K_i = k_4/k_3 = 1$ and two speciation ($\lambda_m$ and $\lambda_{\text{other}}$) and extinction rates ($\mu_m$ and $\mu_{\text{other}}$) that are different between two different karyotype groups as M2 model.

Mapping karyotype data on phylogeny can have a sampling bias that can affect the distribution of species and, hence, speciation or extinction rates between karyotypes. To correct this bias in the M2 and M3 models, we used a correction method in make.musse with inclusion of the sampling fraction, which represents the number of species mapped on the phylogeny divided by the number of species karyotyped. Because we are concerned about the effect of sampling bias on speciation and extinction rates, the average of the sampling fraction for each karyotype state with the same extinction and speciation rate in the model was used as an input value for the sampling fraction. This correction method was not used for the M0, M1, or M4 models. However, for only the likelihood ratio test, which required nested models, we used the same sampling fraction for the M0 model as that used for the M2 model. This M0 model is denoted as the M0' model.

## Reconstruction of ancestral states and statistics of tree branches

Marginal reconstruction of ancestral karyotypes in each node was conducted using asr.marginal function in the diversitree package. To characterize karyotype evolution on each tree branch, we fitted branch length to a given transition of karyotypes using maximized likelihood estimation of the branch length using a likelihood function formulated with the estimated

parameters with the MuSSE method using the M2 model. The transition at each branch was set to the transition from the mode of reconstructed states of the ancestral node of the branch to the mode of reconstructed states of the descendent node of the branch. If the modes of the two nodes are identical, the branch length was set to 0. For each unique transition of all branches, the branch length was estimated by a maximum likelihood method using the mle function in the R package stats4 with the likelihood function described below. Prior for the maximum likelihood estimation was given as 10. To make the likelihood function, we constructed a transition matrix, $Q$, with the parameters $k_1$, $k_2$, $k_3$, and $k_4$ estimated in the M2 model (Table 2). Likelihood of a branch length ($t$) is given by the entry of the matrix exponential, $e^{Qt}$. To reduce the processing time, we used the approximation of likelihood function to 170th-degree polynomial of branch length with the coefficients of the Taylor series of the exponential. This approximation is inaccurate when the branch lengths is too long. Therefore, we designed the likelihood function as follows: if $0 \leq t < 10$, the approximation was used; if $10 < t \leq 15,000$, the matrix exponential was calculated via expm function in an R package, expm (https://cran.r-project.org/web/packages/expm/index.html); if $t < 0$ or $t > 15,000$, "NA" was produced. To confirm the accuracy of the approximation, the estimated maximum likelihood by the approximation was recalculated with the expm function, and we confirmed that the difference of log-likelihood was less than $3.43 \times 10^{-11}$ in the maximum likelihoods.

To classify the karyotype evolution on tree branches statistically, we categorized tree branches into four categories; "Expected," "Conservative," "Rapid," and "Unusual" (S7 Fig). When transitions were within the range expected from the actual branch length ($t_g$) in the ultrametric phylogeny, the transitions were categorized as "Expected." When transitions were out of the range expected from the actual branch length but within the range expected from the fitted branch length calculated above ($t_f$), the transitions were categorized as either "Conservative" or "Rapid." In the case when the fitted branch length was longer than the actual branch length ($t_f > t_g$), the transitions were categorized as "Rapid." When $t_g > t_f$, they were categorized as "Conservative." When transitions were out of the range expected from both the actual and the fitted branch lengths, the transitions were categorized as "Unusual."

To determine whether transitions were within the expected range, we first calculated probabilities of all states calculated in $i$th row of the matrix exponential, $e^{Qt}$, where $i$ is the number of a given initial state. The actual branch length ($t_g$) or the fitted blanch length ($t_f$) were used to calculate the probabilities as explained above. Next, to define the range out of the expected, we calculated the probability of each state and summed all the probabilities from the state with the smallest probability until the cumulative probability (i. e. $p$-value) reached 1%. Any region with a cumulative probability of less than 1% was defined as the unexpected range.

## Phylogenetic comparative analysis of Brassicaceae karyotype evolution with polyploidization being taken into consideration

We generated the data of an ultrametric Brassicaceae phylogenetic tree according to a chronogram report based on the combined data of ndhF/PHYA genes (Supplementary Figure S4 in [57]). We searched for karyotype data of each species on this phylogenetic tree and obtained the karyotype data of 39 species with both chromosome and arm numbers being reported or countable from the figures (S5 Table). For 10 species, possible polyploidization was reported in the original literature (S5 Table). We conducted analyses using either all 39 species or 29 species, in which the 10 reported polyploid species were excluded. We set the limit of the chromosome number $y_{max}$ to 25, which included all the species analyzed. We constructed a new model including polyploidization, which is denoted as the M4 model. The M4 model includes seven free parameters; four evolutionary parameters, $k_1$, $k_2$, $k_3$, and $k_4$, constant speciation and

extinction rates, and an additional parameter for polyploidization, $k_5$. Polyploidization was considered to bring a karyotype from $(x, y)$ to $(2x, 2y)$ at a constant transition rate ($k_5$). The same transition rate was applied for all karyotypes with chromosome number $\leq y_{max}/2$. We also attempted to use the M0 model in Brassicaceae species, but the find.mle function could not complete maximum likelihood estimation within one week. Ancestral state reconstruction was performed using the asr.marginal function. Branches that increased both chromosome number and arm number more than 1.4 fold were considered to have experienced polyploidization in this study.

## Supporting information

**S1 Appendix. Derivation of stationary distribution.**
(DOCX)

**S1 Fig. Comparison between true values and estimates using a karyotype dataset simulated with randomly selected parameters.** In each of the 100 simulation trials, logarithms of $k_1$, $k_2$, $k_3$, and $k_4$ to base 10 were randomly selected from −4 to −1 and used to generate the simulated karyotype dataset. Next, using the simulated datasets, the parameters were estimated using two methods, the MuSSE (M0 model) and Mk-n methods. The first two columns show plots for a true value versus an estimate of each parameter in 100 trials. The third column shows the correlations between two estimates using MuSSE and Mk-n. Pearson's correlation coefficients ($r$) are shown for each panel.
(EPS)

**S2 Fig. Accuracy of parameter estimation using simulated karyotypes and fixed parameters.** Simulations of karyotype evolution were performed separately using three different trees: Eurypterygii, $N = 815$ species; Cyprinodontiformes, $N = 80$ species; Goodeidae, $N = 29$ species. For each tree, 100 simulation trials were performed using four fixed parameters estimated in the following analysis of Eurypterygii. The boxplots show quartiles of $\log_{10}$ fold difference of estimates from the given parameters. The means are shown by the lines in the boxes. The whiskers indicate 1.5 times the interquartile range.
(EPS)

**S3 Fig. The probabilities of the ancestral states of the most recent common ancestors (MRCA) estimated in the simulations: representative results.** Three randomly chosen results for each of the simulations using the three trees (Eurypterygii, Cyprinodontiformes, and Goodeidae) are shown. Plot sizes reflect the probabilities. Plot colors indicate the areas where the top 50%, 70%, 90%, and 100% are included. The 50% and 70% areas overlapped in the first case of Goodeidae.
(EPS)

**S4 Fig. The probabilities of the ancestral states of the most recent common ancestors (MRCA) estimated in the simulations: means of 100 simulations.** The mean probabilities of the ancestral states of the MRCA across 100 simulations are indicated in plot sizes. The plot colors show the areas where the top 50%, 70%, 90%, and 100% are included.
(EPS)

**S5 Fig. Karyograph of all teleost fishes.** Karyotypes of 2,557 species of teleost fish were plotted on the karyograph. The size of the circle indicates $\log_{10}$ of the species number plus one. Thirty outlier species are not plotted here.
(EPS)

**S6 Fig. Karyograph of all teleost fishes with species numbers.** Karyotypes of 2,557 species of teleost fish were plotted on the karyograph. The species number is shown without the log-transformation. Thirty outlier species are not plotted here.
(EPS)

**S7 Fig. Histogram of the frequency of acrocentric chromosomes of teleost fish.** Data of (A) all teleost species ($N$ = 2,587), (B) Eurypterygii only ($N$ = 1,368) and, (C) Otophysi only ($N$ = 1,030) are shown separately.
(EPS)

**S8 Fig. Karyograph of Eurypterygii and Otophysi.** Karyotypes of 1,360 species of Eurypterygii (A) and 1,026 species of Otophysi (B) are plotted on the karyograph. The species number is shown within the dot. Eight outlier species of Eurypterygii and four outlier species of Otophysi are not plotted here.
(EPS)

**S9 Fig. Karyograph showing the probability of ancestral states at three nodes of Eurypterygii inferred with the M0 model.** The ancestral state reconstruction of the MRCA of three clades (Eurypterygii, Cyprinodontiformes, and Goodeidae) is shown. The probability of the ancestral states is indicated by the plot. Plot colors show the areas where the top 50%, 70%, 90%, and 100% are included. The 50% and 70% areas overlapped in Goodeidae.
(EPS)

**S10 Fig. Karyograph showing the probability of ancestral states of the MRCA of Otophysi inferred with the M0 model.** The probability of the ancestral states of MRCA of Otophysi is indicated by the plot sizes. Plot colors show the areas where the top 50%, 70%, 90%, and 100% are included.
(EPS)

**S11 Fig. Simulation analysis to test the enrichment of Eurypterygii with the karyotype (24, 24).** A. Histogram of the percentage of species with karyotype **(24, 24)**. The bin width is 0.12% (1/815). Probability was calculated by 10,000 simulations of karyotype evolution with the estimated parameters in Eurypterygii with the M0 model. B. Karyograph showing the mean proportion of karyotype states. The mean proportion of karyotype states across 10,000 simulations is indicated by the plot sizes. Plot colors show the areas where the top 50%, 70%, 90%, and 100% are included.
(EPS)

**S12 Fig. Karyograph showing the probability of ancestral states at three nodes of Eurypterygii inferred with M2 model.** The ancestral state reconstruction of the MRCA of three clades (Eurypterygii, Cyprinodontiformes, and Goodeidae) is shown. The probability of ancestral states is indicated by plot sizes. Plot colors show the areas where the top 50%, 70%, 90%, and 100% are included. All 50%, 70%, and 90% of the areas overlapped in the three panels.
(EPS)

**S13 Fig. Karyograph showing the probability of ancestral states of the MRCA of Otophysi with the M2 model.** The probability of the ancestral states of the MRCA of Otophysi is indicated by the plot sizes. Plot colors show the areas where the top 50%, 70%, 90%, and 100% are included.
(EPS)

**S14 Fig. Classification of karyotype transitions based on branch length.** The filled circles indicate expected ranges (99%). When the karyotype after the transition is within the range expected from $t_g$, the branch is categorized as "Expected." When the karyotype after the

transition is out of the range expected from $t_g$ but within the range expected from $t_f$, the branch is categorized as "Conservative" ($t_g > t_f$) or "Rapid" ($t_g < t_f$). When the karyotype after the transition is out of the range expected from both $t_g$ and $t_f$, the branch is categorized as "Unusual." $t_g$, actual time estimated from the phylogenetic tree; $t_f$, fitted time of the branch.
(TIF)

**S15 Fig. Karyograph of Brassicaceae species.** Karyotypes of 39 species of Brassicaceae species were plotted on a karyograph. The species number is shown without log transformation.
(EPS)

**S16 Fig. Phylogenetic tree of Brassicaceae with ancestral state reconstruction of the karyotypes.** The circles at the nodes indicate heat maps of mean values of chromosome numbers (A, C) and arm numbers (B, D) in the marginal ancestral reconstruction of karyotypes inferred with the M4 model. The colored points on the tips indicate the karyotypes of the extant species. Black triangles indicate possible polyploidization events. In C and D, analysis was performed after excluding species that were inferred to be polyploid in the original literature.
(EPS)

**S17 Fig. Karyograph showing the probability of ancestral states at the two oldest nodes of the Brassicaceae.** The ancestral state reconstruction of the two oldest nodes of Brassicaceae is shown. The probability of ancestral states is indicated by the plot sizes. Plot colors show the areas where the top 50%, 70%, 90%, and 100% are included.
(EPS)

**S1 Table. Species numbers and karyotype distributions in teleosts shown for each order.**
(XLSX)

**S2 Table. Comparison of maximum likelihood estimates of evolutionary parameters between the model with and without variation of speciation and extinction rates.**
(XLSX)

**S3 Table. Category of karyotype evolution on branches of fish phylogeny.**
(XLSX)

**S4 Table. Top 10 branches with the longest fitted branch length.**
(XLSX)

**S5 Table. Karyotypes of Brassicaceae species.**
(XLSX)

**S6 Table. Maximum likelihood estimates of evolutionary parameters of Brassicaceae.**
(XLSX)

**S7 Table. Maximum likelihood estimates of evolutionary parameters using different limits of chromosome number in fishes.**
(XLSX)

## Acknowledgments

We thank Rumi Suzuki for help with data compilation and all members of Kitano Lab for discussion.

## Author Contributions

**Conceptualization:** Kohta Yoshida, Jun Kitano.

**Data curation:** Kohta Yoshida, Jun Kitano.

**Formal analysis:** Kohta Yoshida.

**Funding acquisition:** Jun Kitano.

**Investigation:** Kohta Yoshida.

**Methodology:** Kohta Yoshida.

**Project administration:** Jun Kitano.

**Software:** Kohta Yoshida.

**Supervision:** Jun Kitano.

**Validation:** Kohta Yoshida, Jun Kitano.

**Visualization:** Kohta Yoshida.

**Writing – original draft:** Kohta Yoshida, Jun Kitano.

**Writing – review & editing:** Kohta Yoshida, Jun Kitano.

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
