## [Decision Letter · Decision Letter 0]

2 Oct 2020

Dear Dr Kitano,

Thank you very much for submitting your Research Article entitled 'Tempo and mode in karyotype evolution revealed by a probabilistic model incorporating both chromosome number and morphology' to PLOS Genetics. Your manuscript was fully evaluated at the editorial level and by independent peer reviewers. The reviewers appreciated the attention to an important problem, but raised some substantial concerns about the current manuscript. Based on the reviews, we will not be able to accept this version of the manuscript, but we would be willing to review again a much-revised version. We cannot, of course, promise publication at that time.

If you decide to revise the manuscript for further consideration at PLOS Genetics, please aim to resubmit within the next 60 days, unless it will take extra time to address the concerns of the reviewers, in which case we would appreciate an expected resubmission date by email to plosgenetics@plos.org.

[LINK]

We are sorry that we cannot be more positive about your manuscript at this stage. Please do not hesitate to contact us if you have any concerns or questions.

Yours sincerely,

Alex Buerkle

Associate Editor

PLOS Genetics

Bret Payseur

Section Editor: Evolution

PLOS Genetics

This manuscript has been carefully evaluated by two referees. Each is enthusiastic about the topic and modeling approach. Both raise key questions about the generality of the method and offer suggestions for comparisons to other modeling approaches and for testing model performance with simulated data. Relatedly, the reviewers seek additional details for how the data selection procedure might have influenced the findings. Given the potential high level of interest in this work and the potential to apply this model to other taxa, the manuscript would benefit from additional attention to these points and others raised in the reviews.

Reviewer's Responses to Questions

**Comments to the Authors:**

Reviewer #1: Yoshida and Kitano present a comprehensive manuscript regarding karyotype evolution reconstruction. Indeed, this is an important and intriguing topic in evolutionary genomics allowing the understanding of how evolution shaped the modern karyotypes in line with adaptation and speciation. In this context, the authors built an original probabilistic model of karyotype evolution incorporating both chromosome and arm numbers, in order to reconstruct the karyotype evolution from common ancestors to modern species in terms of chromosome (numbers and types) and arm numbers. Then, they applied the established model to two major fish taxa, Eurypterygii, and series Otophysi. With this analysis, the authors identified that Eurypterygii species exhibit a conservative karyotype evolution with a high extinction rate in species with a different karyotype while the karyotype evolved in the same direction in multiple lineages of Otophysi making the most common extant karyotype different from the ancestral state.

Overall, I find this manuscript clear and interesting, and such a probabilistic model of karyotype evolution taking into consideration chromosome and arm numbers is indeed novel and represents an asset for karyotype evolution studies. However, I think that some important aspects are still missing in the model, it seems necessary to me to apply it to synteny-based karyotype evolution analyses and reconstructions in plants and/or vertebrates from the literature. It is great to apply the model to two large taxonomic groups of fish and the obtained results are meaningful but don’t look sufficient to assess the robustness of the model and don’t represent significant advances in our understanding of genetic and biological mechanisms. For these reasons, I would classify this manuscript as a method article rather than an original research. Moreover, it still needs to be applied to other taxa and compared with synteny-based analyses, hence the following major issues.

Major issues:

-It would be crucial to apply the established model to other group of species (plants and/or vertebrates) for which synteny-based approaches have been applied to study the karyotype evolution from reconstructed ancestors to the modern species. Indeed, it would be crucial to check whether similar results are obtained and how complementary those two approaches are. For instance, this kind of analysis has been performed in cereals (Murat et al. 2014, GBE) and vertebrates (Simakov et al. 2020, Nature Ecology and Evolution; Nakatani et al. 2007, Genome Research).

-Whole genome duplications (WGD) are a recurrent and major source of karyotype plasticity during plant evolution and also occurred during vertebrate evolution (1R, 2R, 3R). This mechanism impacts dramatically the karyotype during evolution by doubling the chromosome number and has been shown to be followed by extensive chromosomal rearrangements (chromosome number reduction). This parameter should be taken into consideration in the model given that such a mechanism would dramatically bias the outcome of it.

Minor issues:

-Particular attention could be paid to the figure design. For instance, Figure 3 and Figure 4 look very raw and the article would benefit from more sophisticated figures.

-The synteny-based approaches for karyotype evolution studies should be mentioned and referenced.

Reviewer #2: This paper develops and applies a new model for the evolution of genome structure (chromosome number and arm number) and applies it to two fish groups. I think this paper is, on the whole, a great contribution and that the findings are broadly interesting. I believe a few things should probably be explored a bit more, which is discussed below.

Major Comments

line 34-37 (and other parts of the manuscript):

The ending of the abstract and the paper as a whole makes a bold inference with regard to the finding that higher extinction rates are associated with rare karyotypes. However, there is no discussion of the well-documented problem of false positives with this type of SSE approach. (Rabosky and Goldberg 2015, Louca and Pennell 2020, etc.)

I am also curious whether or not the data selection process could be driving this result. In lines 413-415, you describe your selection process; however, you do not report the number of records discarded due to these heuristic selection rules. Likewise, on line 469 you say that you didn't use species with chromosome number greater than 30. How many taxa does this remove from your dataset? Isn't it possible that this thinning of records, particularly extreme ones, could lead to an association with a signal for extinction and extreme karyotypes?

I think there are several things that you could do to get around these problems. One avenue would be to simulate karyographs with your model and inferred parameters on the empirical tree and ask how often you see a result as extreme as what you observe. This would seem to tell you whether you perhaps have a small part of your tree with a strong signal and that any karyotype inferred to be present at that point in the tree will be associated with high extinction rates. Alternatively, you could find that the signal for high extinction rates is supported be many subtrees in your phylogeny and you rarely see a signal as strong as you observe in your data.

Another possible route to confirm or refute this would be to fit a model like BAMM on the full fish tree and your pruned tree. If the rates estimated in the regions where you find higher extinction rates remain similar on the two trees, then your sampling process is unlikely to be causative in your finding.

More broadly, some assessment of the amount of your tree and the number of independent regions of your phylogeny leading to this inference is necessary. What is the effective sample size that is driving this result? By this I don't mean a specific number but how many portions of your tree are supporting this association (i.e. Maddison and Fitzjohn 2015).

Methods section:

Since this is a brand new model of trait evolution, it would be nice to see some assessment of how well it works with simulated data and how well it captures the key aspects of trait evolution in your particular dataset. Perhaps both of these could be accomplished with a posterior predictive simulation (PPS) analysis where you simulate new karyotypes based on the inferred rate parameters and ancestral state inferred at the tree's base. For each of the simulated datasets you could evaluate how well you do at inferring the now known ancestral states as well as the parameters of your model. I think this is essential to know if there are any aspects of the model performance to be concerned with or if certain parameters are difficult to estimate or biased in certain ways.

Line 334: You say that you exclude polyploid species from your analysis. How was this done? Polyploid species often do not have a clear doubling of chromosome number. I think including the number of species excluded from each clade based on these decisions is important. This issue ties in with another concern that your model doesn't allow for chromosome number changes through ascending disploidy. The PPS analysis suggested above might reveal whether this is anything to worry about. For instance, if it is common in your dataset, your simulated datasets will likely show a different chromosome/arm number distribution relative to your empirical dataset.

Minor Comments:

line 42-43: I think that it is inaccurate to say that we don't know if the karyotype itself has any functional role. It seems that a great deal of work has been done considering how chromosome number and the division of the genome into autosomes and sex chromosomes impacts: evolution of certain genetic architectures, geneflow, recombination, and reproductive barriers.

Kitano, J., Ross, J.A., Mori, S., Kume, M., Jones, F.C., Chan, Y.F., Absher, D.M., Grimwood, J., Schmutz, J., Myers, R.M. and Kingsley, D.M., 2009. A role for a neo-sex chromosome in stickleback speciation. Nature, 461(7267), pp.1079-1083.

Anderson, N.W. and Blackmon, H., 2020. The Probability of Fusions Joining Sex Chromosomes and Autosomes. bioRxiv.

line 60-61 I would say that heterokaryotypic individuals experience the full range of fertility effects for instance:

Ratomponirina C, Brun B, Rumpler Y. Kew Chromosome Conference III: Synaptonemal complexes in Robertsonian translocation heterozygous in lemurs.; 1988. p. 65–73

In this study, a range of crosses were made with varying degrees of difference in chromosome number, and spermatogenesis was studied in each F1.

Line 198: I would remind your reader what k1-4 are at this point, it is a lot to keep it in their head and not have to flip back and forth.

Line 234-241: It would be good to include some measures of certainty in the estimates that you report in this paragraph and elsewhere. Saying the most probable is 24 is at best an incomplete picture it could be with 99.9% certainty or all values between 10 and 30 could have nearly identical support, and 24 is just .1% more probable.

Line 506-507: Since diversitree has the sampling proportion approach built into it to deal with sampling biases, why did you not use it? Is your method better or equivalent in performance? If possible, give a citation and or an explanation as to why this choice was made.

**Have all data underlying the figures and results presented in the manuscript been provided?**

Reviewer #1: Yes

Reviewer #2: Yes

PLOS authors have the option to publish the peer review history of their article (what does this mean?). If published, this will include your full peer review and any attached files.

Reviewer #1: No

Reviewer #2: No

---

## [Decision Letter · Decision Letter 1]

22 Mar 2021

Dear Dr Kitano,

We are pleased to inform you that your manuscript entitled "Tempo and mode in karyotype evolution revealed by a probabilistic model incorporating both chromosome number and morphology" has been editorially accepted for publication in PLOS Genetics. Congratulations!

Yours sincerely,

Alex Buerkle

Associate Editor

PLOS Genetics

Bret Payseur

Section Editor: Evolution

PLOS Genetics

Comments from the reviewers (if applicable):

This revised manuscript has been evaluated by two referees, both of whom praise the authors' response to the previous round of review. I appreciate the authors' attention to the reviewer suggestions and the additions and changes that were made to the manuscript.

Reviewer's Responses to Questions

**Comments to the Authors:**

Reviewer #1: Yoshida and Kitano made a great effort to address the issues I raised by taking into consideration polyploidization events in their model and by applying it to a group of plants (Brassicaceae). Their results make sense compared to synteny-based analyses and their model is now even more accurate to study karyotype evolution in various groups of species. Therefore, they answered all my concerns. Congratulations!

Reviewer #2: I'm pleased with the author's response to my comments. I appreciate the time and effort spent to include simulation analyses to give some idea of performance of the method. One small text note on line 56 it seems that you could just end your sentence with the parenthetical statement and delete "of the chromosome."

**Have all data underlying the figures and results presented in the manuscript been provided?**

Reviewer #1: Yes

Reviewer #2: Yes

PLOS authors have the option to publish the peer review history of their article (what does this mean?). If published, this will include your full peer review and any attached files.

Reviewer #1: No

Reviewer #2: No

**Data Deposition**

http://datadryad.org/submit?journalID=pgenetics&manu=PGENETICS-D-20-01321R1

**Press Queries**

---

## [Editor Report · Acceptance letter]

12 Apr 2021

PGENETICS-D-20-01321R1 

Tempo and mode in karyotype evolution revealed by a probabilistic model incorporating both chromosome number and morphology 

Dear Dr Kitano, 

We are pleased to inform you that your manuscript entitled "Tempo and mode in karyotype evolution revealed by a probabilistic model incorporating both chromosome number and morphology" has been formally accepted for publication in PLOS Genetics! Your manuscript is now with our production department and you will be notified of the publication date in due course.

With kind regards,

Andrea Szabo

PLOS Genetics

On behalf of:
